# Characterization of Inhibitory Effectiveness in Hyperpolarization-Activated Cation Currents by a Group of *ent*-Kaurane-Type Diterpenoids from *Croton tonkinensis*

**DOI:** 10.3390/ijms21041268

**Published:** 2020-02-13

**Authors:** Ping-Chung Kuo, Yen-Chin Liu, Yi-Ching Lo, Sheng-Nan Wu

**Affiliations:** 1School of Pharmacy, National Cheng Kung University Medical College, Tainan 70101, Taiwan; z10502016@email.ncku.edu.tw; 2Department of Anesthesiology, National Cheng Kung University Hospital, College of Medicine, National Cheng Kung University, Tainan 70101, Taiwan; inp1965@mail.ncku.edu.tw; 3Department of Pharmacology, Kaohsiung Medical University, Kaohsiung 80708, Taiwan; yichlo@kmu.edu.tw; 4Department of Physiology, National Cheng Kung University Medical College, Tainan 70101, Taiwan; 5Institute of Basic Medical Sciences, National Cheng Kung University Medical College, Tainan 70101, Taiwan

**Keywords:** croton, pituitary cell, pancreatic β-cell, hyperpolarization-activated cation current, *erg*-mediated K^+^ current, membrane potential

## Abstract

Croton is an extensive flowering plant genus in the spurge family, Euphorbiaceae. Three croton compounds with the common *ent*-kaurane skeleton have been purified from *Croton tonkinensis*. Methods: We examined any modifications of croton components (i.e., croton-01 [*ent*-18-acetoxy-7α-hydroxykaur-16-en-15-one], croton-02 [*ent*-7α,14β-dihydroxykaur-16-en-15-one] and croton-03 [*ent*-1β-acetoxy-7α,14β-dihydroxykaur-16-en-15-one] on either hyperpolarization-activated cation current (*I*_h_) or *erg*-mediated K^+^ current identified in pituitary tumor (GH_3_) cells and in rat insulin-secreting (INS-1) cells via patch-clamp methods. Results: Addition of croton-01, croton-02, or croton-03 effectively and differentially depressed *I*_h_ amplitude. Croton-03 (3 μM) shifted the activation curve of *I*_h_ to a more negative potential by approximately 11 mV. The voltage-dependent hysteresis of *I*_h_ was also diminished by croton-03 administration. Croton-03-induced depression of *I*_h_ could not be attenuated by SQ-22536 (10 μM), an inhibitor of adenylate cyclase, but indeed reversed by oxaliplatin (10 μM). The *I*_h_ in INS-1 cells was also depressed effectively by croton-03. Conclusion: Our study highlights the evidence that these *ent*-kaurane diterpenoids might conceivably perturb these ionic currents through which they have high influence on the functional activities of endocrine or neuroendocrine cells.

## 1. Introduction

The genus *croton* (Euphorbiaceae) includes about 300 species that are widely distributed throughout tropical regions. *C. tonkinensis* Gagnep. (Euphorbiaceae), known as *Kho sam cho la* in Vietnamese, is a tropical shrub native to northern Vietnam and has been used in Vietnam for the treatment of various types of disorders [1]. Previous phytochemical investigations have shown that *C. tonkinensis* is a rich source of diterpenoids [1]. Anti-inflammatory and cancer chemopreventive activities of *C. tonkinensis* extracts have been recently reported. Those effects are thought to be linked to their ability to depress the transcription nuclear factor κB [1].

Previous studies have demonstrated that intravenous injection with the essential oil of *C. zehntneri* or *nepetaefolius* could induce rapid and dose-dependent hypotension and bradycardia [2,3,4]. The essential oil of *C. nepetaefolius* was reported to reverse histamine-induced firing in guinea-pig celiac ganglion [5]. The diterpene isolated from *C. argyrophylloides* was demonstrated to exert antispasmodic effects in airway smooth muscle [6].

Earlier reports have shown that the compounds purified from croton could modulate different types of ion channels. For example, either crofelemer or SP-300, a purified proanthocyanidine oligomer extracted from the South American medicinal plant *C. lechleri* (dragon’s blood), has been demonstrated to depress Cl^-^ currents in colonic epithelial cells [7,8]. The diterpene isolated from *C. argyrophylloides* could block voltage-gated Ca^2+^ channels in airway smooth muscle cells [6].

Hyperpolarization-activated cation current (*I*_h_) has been recognized as a key determinant of repetitive electrical activity present in heart cells and in a variety of central neurons, and neuroendocrine or endocrine cells [9,10,11,12,13,14,15,16,17]. This current with the slow activation kinetics is a mixed inward Na^+^/K^+^ current which is sensitive to block by CsCl, ivabradine or zatebradine [18,19], and the increase in this current magnitude can result in depolarizing membrane potential to threshold required for the generation of action potential (AP), thereby influencing pacemaker activity and impulse propagation [10,13,16,20,21]. Correspondingly, the slow kinetics of *I*_h_ in response to long-step membrane hyperpolarization can also produce long-lasting activity-dependent changes in the excitability of electrically excitable cells [17]. It has been demonstrated to be carried by the channels encoded by a family of the hyperpolarization-activated cyclic nucleotide-gated (HCN) genes, which belongs to the superfamily of voltage-gated K^+^ channels and cyclic nucleotide-gated channels [10]. However, whether or how croton or other *ent*-kaurane-type diterpenoids can interact with these HCN channels to modify the amplitude and gating of *I*_h_ as well as to perturb membrane potential has thus far remained unexplored.

Therefore, the objective of this work was to investigate whether the fractions purified from *C. tonkinensis*, namely croton-01 (*ent*-18-acetoxy-7α-hydroxykaur-16-en-15-one), croton-02 (*ent*-7α,14β-dihydroxykaur-16-en15-one) and croton-03 (*ent*-1β-acetoxy-7α,14β-dihydroxykaur-16-en-15-one), could exert any perturbations on the amplitude and gating of *I*_h_ in pituitary GH_3_ cells and in INS-1 insulin-secreting cells. Of interest, findings from the present results provide the evidence to unravel that these compounds can effectively interact with the HCN channel to inhibit *I*_h_ in a concentration-, time-, and state-dependent manner. Due to croton-03′s high potency, the present study might be important in evaluating the in vivo mechanisms through which croton-03 or other *ent*-kaurane-type diterpenoids produce cellular functions (e.g., antinociceptive action) [22,23].

## 2. Results

### 2.1. Effects of Croton-03 and Other ent-Kaurane Diterpenoids on Hyperpolarization-Activated Cation Current (I_h_) Identified in Pituitary GH_3_ Cells

In an initial stage of experiments, the modifications of *I*_h_ caused by croton-03 were evaluated in GH_3_ cells which were immersed in Ca^2+^-free Tyrode’s solution, and the recording pipette used was filled with K^+^-containing solution. As the whole-cell current recordings were firmly established, we maintained the examined cells at the level of −40 mV, and a long-lasting membrane hyperpolarization to −110 mV was thereafter applied. Under this experimental protocol, a slowly activating current elicited by sustained hyperpolarization was readily evoked and has hence been referred as an *I*_h_ [14,18,24]. Notably, as cells were exposed to croton-03, the amplitudes of *I*_h_ by maintained hyperpolarization was progressively diminished together with a considerable slowing in activation time course of the current (Figure 1A). For example, cell exposure to 3 μM croton-03 decreased current amplitude at the end of hyperpolarizing pulse from 309±21 to 168 ± 17 pA (*n* = 8, *p* < 0.05). After washout of the agent, current amplitude returned to 287 ± 19 pA (*n* = 8). In addition to the decreased *I*_h_ amplitude, the value of activation time constant (τ_act_) of the current in response to maintained hyperpolarization was evidently raised as cells were exposed to croton-03 (Figure 1B,C). For example, the presence of 3 μM croton-03 was able to increase the τ_act_ value to 1158 ± 113 msec (*n* = 8, *p* < 0.05) from a control value of 567 ± 54 msec (*n* = 8).

The effects of these diterpenoids (i.e., croton-01, croton-02 and croton-03) on *I*_h_ in response to membrane hyperpolarization were further examined and compared in this study. The concentration-dependent relationships among the inhibitory effects of these agents on *I*_h_ are illustrated in Figure 1D. The IC_50_ value of croton-01, croton-02, or croton-03 that were required for the inhibition of *I*_h_ amplitude was calculated to be 2.89, 6.25, or 2.84 μM, respectively; however, the Hill coefficients obtained among their effects on *I*_h_ did not differ significantly. Therefore, croton-01 or croton-03 tends to be more potent in depressing *I*_h_ than croton-02.

### 2.2. Effects of Croton-03 on I-V Relationship and Steady-State Activation Curve of I_h_ in GH_3_ Cells

We further examined the effect of croton-03 on *I*_h_ taken at different levels of membrane potentials. The *I*_h_ was elicited when the cell was hyperpolarized from −40 to a series of voltage steps ranging between −110 and −30 mV (Figure 2B). Croton-03 at a concentration of 3 μM was able to depress the amplitude of *I*_h_ measured at the end of each hyperpolarizing step. The overall *I-V* relationships of *I*_h_ with or without addition of croton-03 are shown in Figure 2B. Figure 2C shows the steady-state activation curve of *I*_h_ taken with or without addition of croton-03 (3 μM). In control (i.e., in the absence of croton-03), *V*_1/2_ = −96.2 ± 2.6 mV, *q* = 3.2 ± 0.8 *e* (*n* = 7), while in the presence of 3 μM croton-03, *V*_1/2_ = −84.3 ± 2.4 mV, *q* = 3.3 ± 0.7 *e* (*n* = 7). Therefore, it is clear from the present results that croton-03 not only led to a conceivable reduction in *I*_h_ magnitude, but it also significantly shifted the activation curve along the voltage axis to a more negative potential by approximately 11 mV, irrespectively of minimal change in the gating charge (*q*) of the curve.

### 2.3. Effect of Croton-03 on the Voltage-Dependent Hysteresis of I_h_ Elicited in Response to Long-Lasting Triangular Ramp Pulse

The voltage-dependent hysteresis of ionic currents (e.g., *I*_h_) has been previously demonstrated to exert a significant impact on electrical behaviors such as the firing of action potentials [25,26,27]. For this reason, we further explored whether there is possible voltage-dependent hysteresis existing in *I*_h_ recorded from GH_3_ cells. In this series of experiments, we delivered a long-lasting triangular ramp pulse with a duration of 2 sec (i.e., ±0.11 V/sec), as whole-cell mode was firmly established. It is evident from Figure 3A that the trajectories of *I*_h_ elicited by the upsloping (i.e., depolarizing from −150 to −40 mV) and downsloping (hyperpolarizing from −40 to −150 mV) ramp pulse as a function of time were distinguishable between them. The current amplitude elicited by the upsloping limb of triangular voltage ramp was significantly greater than that by the downsloping limb, indicating that there is a voltage hysteresis for this current as depicted in Figure 3B, namely the relationship of *I*_h_ versus membrane potential. As the ramp speed became reduced, the hysteresis degree for *I*_h_ was greatly raised. Of note, as the examined cell was exposed to croton-03 (3 μM), *I*_h_ amplitude evoked in the upsloping limb of long-lasting triangular ramp was noted to be decreased to a greater extent than that measured from the downsloping limb. For example, in controls, *I*_h_ at the level of -100 mV elicited upon the upsloping and downsloping ends of triangular ramp pulse were 213 ± 23 and 97 ± 12 pA (*n* = 7), respectively, the values of which were found to differ significantly between them (*p* < 0.05). Moreover, in the presence of 3 μM croton-03, the amplitudes of forward and backward *I*_h_ taken at the same level of membrane potential were significantly reduced to 136 ± 16 and 58 ± 9 pA (*n* = 7, *p* < 0.05), respectively. However, the magnitude of croton-03-induced current inhibition at the upsloping and downsloping limbs of triangular ramp did not differ significantly, i.e., the presence of 3 μM croton decreased *I*_h_ amplitude (at −100 mV) in the upsloping and downsloping limb by about 40%.

We next quantified the degree of voltage-dependent hysteresis based on the difference in area under the curve in the forward (upsloping) and reverse (downsloping) direction as described by the arrows in Figure 3B. It was seen that for *I*_h_ identified in GH_3_ cells, the degree of voltage hysteresis increased with slower ramp speed, and that the presence of croton-03 led to a considerable reduction in the amount of such hysteresis. For example, as the duration of triangular ramp was prolonged from 2 to 3 sec, the area under the curve between forward and backward current traces (i.e., Δarea) increased from 23.9 ± 4.2 to 34.1 ± 4.9 mV·pA, (*n* = 7, *p* < 0.05). Figure 3C illustrates summary of the data showing effects of croton-03 at the different concentrations on the area under such curve. For example, apart from its depression of *I*_h_ magnitude, addition of croton-03 (3 μM) significantly decreased the area by about 50% elicited in response to such long-lasting triangular voltage ramp, as demonstrated by a significant decrease of Δarea from 23.9 ± 4.2 to 11.1 ± 2.2 mV·nA (*n* = 7, *p* < 0.05).

### 2.4. Effect of Croton-03, YS-035, Zatebradine, Croton-03 Plus SQ-22536, and Croton-03 Plus Oxaliplatin on I_h_ Amplitude

The extracts from *C. zehntneri* such as anethole and estragole were demonstrated to produce an increase in the level of intracellular cyclic AMP in the copora cavernosa smooth muscle [28]. In the next set of experiments, we examined and compared the effects of croton-03, YS-035, zatebradine, croton-03 plus SQ-22536, and croton-03 plus oxaliplatin on *I*_h_ amplitude in GH_3_ cells. YS-035 or zatebradine was recognized as an inhibitor of *I*_h_ [19,29], oxaliplatin was recently demonstrated to stimulate *I*_h_ [30,31], and SQ-22536 is an inhibitor adenylate cyclase. As illustrated in Figure 4, similar to the effect of croton-03 on *I*_h_ described above, the addition of either YS-053 (3 μM) or zatebradine (3 μM) was effective at depressing *I*_h_ amplitude. Subsequent application of SQ-22536 (10 μM), still in the presence of croton-03, had minimal effect on *I*_h_ produced by croton-03; however, further addition of oxaliplatin (10 μM) significantly attenuated croton-03-mediated inhibition of *I*_h_. Alternatively, chlorotoxin (1 μM), an inhibitor of Cl^-^ channels, did not depress *I*_h_ amplitude effectively (data not shown). Consequently, the experimental results led us to indicate that croton-03-mediated depression of *I*_h_ appears to be independent of its increase in intracellular cyclic AMP, though it is sensitive to be reversed by further addition of oxaliplatin.

### 2.5. Effect of Croton-03 on erg-Mediated K^+^ Current (I_K(erg)_) in GH_3_ Cells

In another set of current recordings, we investigated the possible modification of croton-03 on *I*_K(erg)_. In these experiments, we bathed cells in high-K^+^, Ca^2+^-free solution and, during the measurements, we filled the pipette solution by using K^+^-containing solution. As the whole-cell current recordings were achieved, the examined cell was maintained at the level of -10 mV and a long-step membrane hyperpolarization was applied. As illustrated in Figure 5, the addition of croton-03 at a concentration of 3 or 10 μM slightly depressed the amplitude of deactivating *I*_k(erg)_, the biophysical or pharmacological properties of which were previously described [32]. However, minimal change in the decaying time course of *I*_K(erg)_ in response to membrane hyperpolarization was demonstrated in the presence of croton-03. The concentration-dependent inhibition of croton-03 on *I*_K(erg)_ was then constructed with an IC_50_ value of 35.3 μM, which is higher than that for the inhibition of *I*_h_ described above. As such, the *I*_k(erg)_ identified in GH_3_ is relatively resistant to depression by the presence of croton-03.

### 2.6. Effect of Croton-03 on Sag Potential Measured from GH_3_ Cells

We further performed current-clamp voltage recordings in attempts to test if croton-03 could produce any perturbations on sag potential in these cells. Sag potential in response to hyperpolarizing current stimulus has been described to be intimately linked to the occurrence of *I*_h_ in different types of central neurons [20,33,34]. As shown in Figure 6, under our experimental condition, when the whole-cell voltage recordings were firmly established, a long-step hyperpolarizing current injection with the amplitude of around 25 pA was found to induce sag potential (i.e., drop down to a lower level in the membrane potential upon hyperpolarizing current stimuli). The addition of ivabradine (3 μM), an inhibitor of *I*_h_, was effective at depressing the amplitude of sag potential. Moreover, the presence of 1 or 3 μM croton-03 produced an effective depression of sag potential elicited by hyperpolarizing current stimulus. For example, croton-03 at a concentration of 3 μM decreased the amplitude of sag potential from 29.2 ± 2.1 to 18.3 ± 1.5 mV (*n* = 8, *p* < 0.05). Therefore, the sag potential identified in GH_3_ cells is connected with the magnitude of *I*_h_ and the depression of such potential caused by the presence of croton-03 could result largely from its inhibitory effect on *I*_h_ observed in GH_3_ cells.

### 2.7. Inhibitory Effect of Croton-03 on I_h_ in INS1 Insulin-Secreting Cells

The biophysical or pharmacological properties of *I*_h_ observed in GH_3_ cells may be distinguishable from those in other types of endocrine cells. Previous studies have demonstrated the presence of *I*_h_ or delayed-rectifier K^+^ current in pancreatic β-cells [35,36]. INS-1 cells have retained some characteristics of normal pancreatic β-cells [37]. In another set of whole-cell current recordings, we thus further attempted to test whether croton-03 is capable of perturbing ionic currents (e.g., *I*_h_ or *I*_K(erg)_) present in INS-1 insulin-secreting cells. As illustrated in Figure 7, indistinguishable from the observations made in GH_3_ cells, the presence of croton-03 effectively depressed the amplitude of *I*_h_ in response to long-step membrane hyperpolarization. Moreover, subsequent addition of oxaliplatin (10 μM), still in the presence of croton-03 (3 μM), could significantly attenuate croton-03-mediated inhibition of the *I*_h_ amplitude. Oxaliplatin was previously demonstrated to stimulate *I*_h_ amplitude [30,31].

### 2.8. Effect of Croton-03 on I_K(erg)_ Identified in INS1 Cells

The *I*_K(erg)_ identified in INS-1 cells was also tested to evaluate whether croton-03 has any effect on this current. Cells were bathed in high-K^+^, Ca^2+^-free solution, and the recording pipette was filled with K^+^-containing solution. As illustrated in Figure 8, when the membrane potential was hyperpolarized from −10 to −140 mV, the deactivating *I*_K(erg)_ was readily evoked in these cells. Moreover, within 2 min of exposing to croton-03 at a concentration of 10 or 30 μM, the *I*_k(erg)_ amplitude elicited by step hyperpolarization was robustly reduced. For example, the addition of croton-03 (30 μM) decreased current amplitude measured at the beginning of hyperpolarizing pulse from 984 ± 43 to 713 ± 28 pA (*n* = 8, *p* < 0.05). Hence, in keeping with the observations described above in GH_3_ cells, the *I*_K(erg)_ identified in INS-1 cells was subject to be mildly but significantly depressed by croton-03.

## 3. Discussion

The principal findings in this study were as follows: (1) the presence of croton-01, croton-02, or croton-03 produced concentration-dependent inhibition of *I*_h_ in pituitary GH_3_ cells with effective IC_50_ of 2.89, 6.25 or 2.84 μM, respectively.; (2) croton-03 shifted the steady-state activation curve of *I*_h_ toward a more negative potential; (3) croton-03 decreased the voltage-dependent hysteresis of *I*_h_ and further addition of SQ-22536 did not alter croton-03-mediated inhibition of *I*_h_; (4) croton-03 mildly depressed the amplitude of *I*_K(erg)_; (5) croton-03 decreased the amplitude of sag potential elicited by hyperpolarizing current stimulus; and (6) croton-03 also depressed *I*_h_ amplitude in INS-1 insulin-secreting cells.

A previous report has demonstrated the ability of extracts from *C. zehntneri* (i.e., anethole and estragole) to increase the levels of cyclic AMP in corpora cavernosa smooth muscle [28]. However, in continued presence of croton-03, further addition of SQ-22536, an inhibitor of adenylate cyclase, was unable to attenuate croton-03-mediated depression of *I*_h_ seen in GH_3_ cells. Therefore, croton-03-induced block of *I*_h_ is unlikely to ascribe largely from changes in the level of intracellular cyclic AMP.

In light of the steady-state activation curve of *I*_h_ during the exposure of croton-03, the voltage for half-maximal activation was noted to be in the range of the firing of action potentials. The activation time course of *I*_h_ observed at the different level of voltages became slower in the presence of croton-03. Therefore, it is conceivable that the croton-03 molecule has a higher affinity toward the open state of HCN channels than toward the closed or resting state of the channels identified in GH_3_ or INS-1 cells. As such, the magnitude of croton-03-induced block on *I*_h_ tends to be voltage-dependent, and any changes in *I*_h_ amplitude caused by croton-03 would fairly rely on the croton-03 concentration applied, the firing of action potentials, and the pre-existing level of membrane potential.

The voltage-dependent hysteresis of *I*_h_ has been shown to exhibit substantial role in influencing electrical behavior of electrically excitable cells such as GH_3_ cells. In keeping with previous observations [25,26,27], the *I*_h_ identified in GH_3_ cells was found to undergo either a hysteresis in its voltage dependence, or mode shift in which the voltage sensitivity of gating charge movements relies on the previous state [26,27]. In this study, we also examined possible perturbations of croton-03 on such non-equilibrium property of *I*_h_ in GH_3_ cells. Our results clearly demonstrated that the presence of this diterpenoid (croton-03) was capable of diminishing such hysteresis involved in the voltage-dependent elicitation of *I*_h_.

In this study, the effective IC_50_ of croton-03 required for the inhibition of *I*_h_ seen in GH_3_ cells was 2.84 μM, a value that is similar to that of croton-01, but slightly lower than that of croton-02. This discrepancy remains unclear; however, it could be related to the possibility that croton-03 is assumed to be more hydrophobic than croton-01 or croton-02. The presence of croton-03 was also noted to depress the activation time course of *I*_h_ as well as to shift the activation curve of the current to a more depolarized potential. Regardless of the detailed mechanism of its inhibitory actions, the *I_h_* (or HCN-encoded current) could conceivably be an important target of these diterpenoids such as croton-3, which can interact with the HCN channel to decrease the amplitude of *I*_h_ in endocrine cells such as GH_3_ and INS-1 cells. The present results also imply the possibility of its perturbations on different cell types which express functional HCN channels such as sensory neurons [38,39]. Indeed, our results demonstrated that subsequent addition of oxaliplatin, but still in the presence of croton-03, could significantly attenuate croton-03-mediated inhibition of *I*_h_ in GH_3_ and INS-1 cells. Oxaliplatin, an activator of *I*_h_, was recognized as a chemotherapy-induced neuropathic pain [30,31].

It needs to be mentioned that the IC_50_ value (i.e., 2.84 μM) estimated for croton-03-mediated inhibition of *I*_h_ tends to be so high that it may be reached anywhere in the organism upon consumption of even large amounts of *C*. tonkinensis. Therefore, to what extent the results observed herein have a real physiological, nutrition-related relevance remains to be further investigated.

The *I*_h_ has been recognized to be carried by channels of the HCN gene family, namely HCN1, HCN2, HCN3 and HCN4 [10]. It has been recently demonstrated that HCN1, HCN2 or HCN3 channels play essential roles in sensory excitability and pain sensation [30,38,40]. Regardless of the detailed mechanism of their inhibitory actions, there is likely to be a relevant link between the effects of these croton derivatives on electrically excitable cells and their inhibitory effects on HCN channel activity, though croton-03-mediated effect on HCN channels might not be isoform-specific. It is tempting to speculate that the blocking by these *ent*-kaurane-type diterpenoids of *I*_h_ plays an important role in modulating the functional activities of neurons and neuroendocrine or endocrine cells [5,6]. Therefore, croton derivatives and other structurally related *ent*-kaurane-type diterpenoids would be the harbinger of intriguing compounds that use the open/activated state of the HCN channels as a substrate. Our study strongly suggests that these *ent*-kaurane diterpenoids should somehow act on cellular mechanisms through which they exert high influence on the functional activities of endocrine or neuroendocrine cells.

## 4. Materials and Methods

### 4.1. Purification, Extraction, and Fractionation of Plant Materials

The general procedures follow those shown in the previous research [41]. The whole *C. tonkinensis* Gagnep (Euphorbiaceae) plant has been collected in Vietnam, and the plant material was identified and authenticated by Vu Xuan Phuong, Institute of Ecology and Biological Resources, Vietnamese Academy of Science and Technology, Hanoi, Vietnam. A voucher specimen (Viet-TSWu-2009-0901-001) was deposited in the herbarium of the Institute of Ecology and Biological Resources, Vietnamese Academy of Science and Technology.

Extraction and fractionation were performed as described previously [41]. Briefly, air-dried and powdered whole plants of *C. tonkinensis* were extracted with methanol and concentrated to form brown syrup. The *n*-hexane soluble fraction was subjected to silica gel column chromatography eluted with a step gradient to afford 19 fractions (H1-H19). Minor fractions in subfraction H13-4, H15-2-2 and H15-3-3 were recrystallized with chloroform methanol to yield croton-01, croton-03, and croton-02, respectively.

### 4.2. Drugs, Chemicals, and Solutions

The chemical structures of croton-01 (*ent*-18-acetoxy-7α-hydroxykaur-16-en-15-one), croton-02 (*ent*-7α,14β-dihydroxykaur-16-en-15-one) and croton-03 (*ent*-1β-acetoxy-7α,14β-dihydroxykaur-16-en-15-one) were described previously [41,42]. β-Mercaptoethanol and tetrodotoxin were acquired from Sigma-Aldrich (St. Louis, MO), ivabradine, SQ-22536, YS-035 and zatebradine were from Tocris (Bristol, UK), and oxaliplatin was from Sanofi-Aventis (New York, NY, USA). Chlorotoxin was kindly provided by Professor Dr Woei-Jer Chuang (Department of Biochemistry, National Cheng Kung University Medical College, Tainan, Taiwan). Cell culture media such as Ham’s F-12 or RPMI-1640 medium, fetal bovine or calf serum, and horse serum were acquired from Invitrogen (Carlsbad, CA, USA), unless stated otherwise, while other chemicals or solvents such as methanol, HEPES, and aspartic acid, were of analytical reagent grade. The twice-distilled water that had been de-ionized through a Millipore-Q system (APS Water Services Inc., Van Nuys, CA, USA) was used in all experiments.

The composition of bath solution (i.e., HEPES-buffered normal Tyrode’s solution) was as follows (in mM): NaCl 136.5, KCl 5.4, CaCl_2_ 1.8, MgCl_2_ 0.53, glucose 5.5, and HEPES-NaOH buffer 5.5 (pH 7.4). To measure membrane potential or K^+^ currents (i.e., *I*_K(erg)_) or *I*_h_, we filled the patch pipette with a solution (in mM): KCl 140, MgCl_2_ 1, Na_2_ATP 3, Na_2_GTP 0.1, EGTA 0.1, and HEPES-KOH buffer 5 (pH 7.2). To record *I*_K(erg)_, we bathed GH_3_ or INS-1 cells in high-K^+^, Ca^2+^-free solution (in mM): KCl 145, MgCl_2_ 0.53, and HEPES-KOH 5 (pH 7.4). The pipette solution and culture media were commonly filtered on the day of use with sterile Acrodisc^®^ filter with 0.2-μm Supor^®^ membrane (Pall Corp., Port Washington, NY, USA).

### 4.3. Cell Preparations

GH_3_ pituitary tumor cells, obtained from the Bioresources Collection and Research Center ([BCRC-60015]; Hinchu, Taiwan), were routinely maintained in Ham’s F-12 media supplemented with 15% horse serum (*v*/*v*), 2.5% fetal calf serum (*v*/*v*), and 2 mM L-glutamine [14]. The rat INS-1 cell line (clone 832/13) was kindly provided by Christopher B. Newgard, Duke University, Durham, NC. Cells were plated in 10-cm plate and grown in RPMI-1640 medium supplemented with 11.1 mM D-glucose, 10% fetal bovine serum, 10 mM HEPES, 2 mM L-glutamine, 1 mM sodium pyruvate, and 50 μM β-mercaptoethanol [34]. The experiments were commonly made five or six days after cells had been cultured (60–80% confluence).

### 4.4. Electrophysiological Measurements

Shortly before each experiment, cells (i.e., GH_3_ or INS-1 cells) were dissociated and a few drops of cell suspension was transferred to a home-made chamber mounted on the fixed stage of an inverted Diaphot-200 microscope (Nikon, Tokyo, Japan). They were immersed at room temperature (20–25 °C) in normal Tyrode’s solution, the composition of which is described above. We fabricated the recording electrode from Kimax-51 glass capillaries (#34500; Kimble, Vineland, NJ) using a PP-830 vertical puller (Narishige, Tokyo, Japan) in which a two-step pull mechanism was applied, and their tips were fire-polished with a microforge (MF-83, Narishige). During the measurements, the electrode with tip resistance ranging from 3 to 5 MΩ, which was firmly inserted into holder, was maneuvered by use of a WR-98 micromanipulator (Narishige). Patch-clamp experiments operated under voltage- or clamp-clamp mode were carried out by using an RK-400 patch-clamp amplifier (Bio-Logic, Claix, France) connected with a personal computer [43]. Shortly before giga-seal formation was achieved, the potentials were commonly corrected for the liquid junction potential that generally developed at the pipette tip, as the composition of internal solution was different from that in the bath.

### 4.5. Data Recordings

The data comprising both potential and current traces were collected online and stored in an HP Pavilion ×360 touchscreen laptop computer (14-cd1053TX; Hewlett-Packard, Palo Alto, CA, USA) at 10 kHz equipped with the 12-bit Digidata 1440A interface (Molecular Devices, Sunnyvale, CA, USA). The latter device was widely accepted and used for efficient analog-to-digital/digital-to-analog conversion. During the experiments, the data acquisition system was driven by pCLAMP 10.7 software (Molecular Devices) run under Windows 10 (Redmond, WA, USA), and the current/voltage signals were also simultaneously monitored on LCD monitor (MB169B+; ASUS, Taipei, Taiwan) through a USB type-C connection. Current signals were low-pass filtered at 2 kHz with an FL-4 four-pole Bessel filter (Dagan, Minneapolis, MN, USA). Through digital-to-analog conversion, the pCLAMP-generated voltage-clamp patterns with different rectangular, triangular or linear ramp waveforms were designed beforehand, in attempts to determine the current-voltage (*I-V*) relationships for different types of ionic currents such as *I*_h_, and the steady-state activation curve of *I*_h_.

### 4.6. Data Analyses

To determine percentage inhibition of croton-01, croton-02, or croton-03 on *I*_h_, each cell was hyperpolarized from −40 to −120 with a duration of 2 s, while to evaluate that of croton-03 on *I*_K(erg)_, the cell was hyperpolarized from −10 to −90 mV. The *I*_h_ or *I*_K(erg)_ amplitudes respectively measured at the end or beginning of each hyperpolarizing step in the presence of different concentrations of compounds were compared with the control value (i.e., in the absence of any agent). The concentration-dependent effects of these compounds on the inhibition of *I*_h_ or *I*_K(erg)_ amplitude seen in GH_3_ cells were fitted with the goodness of fit to a modified Hill function by 64-bit OriginPro 2016 (OriginLab); that is,
(1)$$\mathrm{Percentage}\;\mathrm{inhibition}\;\left(\%\right)=\frac{E_{max}\times {\left[C\right]}^{n_{H}}}{{\left[C\right]}^{n_{H}}+IC_{50}^{n_{H}}}$$
where [C] is the concentration of croton-01, croton-2, or croton-03; IC_50_ and n_H_ are the concentration required for a 50% inhibition and the Hill coefficient, respectively; and *E*_max_ is the maximal inhibitions of *I*_h_ or *I*_k(erg)_ amplitude produced by each tested compound.

To evaluate the effect of croton-03 on the steady-state activation of *I*_h_, the relationships between the normalized amplitude of *I*_h_ and the conditioning potentials obtained in the absence and presence of 3 μM croton-03 were least-squares fitted with a Boltzmann function of the following form:(2)$$I=\frac{I_{max}}{1+exp\left[\frac{-\left(V-V_{\frac{1}{2}}\right)qF}{RT}\right]}$$
where *I*_max_ is the maximal amplitude of *I*_h_ taken with or without addition of 3 μM croton-03 at the conditioning potential of −30 mV, *V*_1/2_ the voltage at which half-maximal activation occurs, *q* the apparent gating charge, *F* Faraday’s constant, *R* the universal gas constant, and *T* the absolute temperature.

### 4.7. Statistical Analyses

Linear or nonlinear curve fitting to data shown here was performed by using either OriginPro 2016 (OriginLab, Northampton, MA) or the Solver subroutine embedded in Excel 2016 (Microsoft) [44]. The data are presented as mean±standard errors of the mean (SEM) and error bars were plotted as SEM. Values of *n* indicate the number of cells obtained. The paired or unpaired Student’s *t*-test and one-way analysis of variance (ANOVA) followed by the least significance difference method for multiple comparisons were used for the statistical evaluation of differences among means. Non-parametric Kruskal-Wallis test would be implemented, as the assumption of normality underlying ANOVA was violated. Statistical analyses were generally made using Statistical Package for the Social Sciences version 20 (SPSS, IBM Corp., Armonk, NY, USA). The level of statistical significance was set at *p* < 0.05.

## Figures and Tables

**Figure 1 ijms-21-01268-f001:**
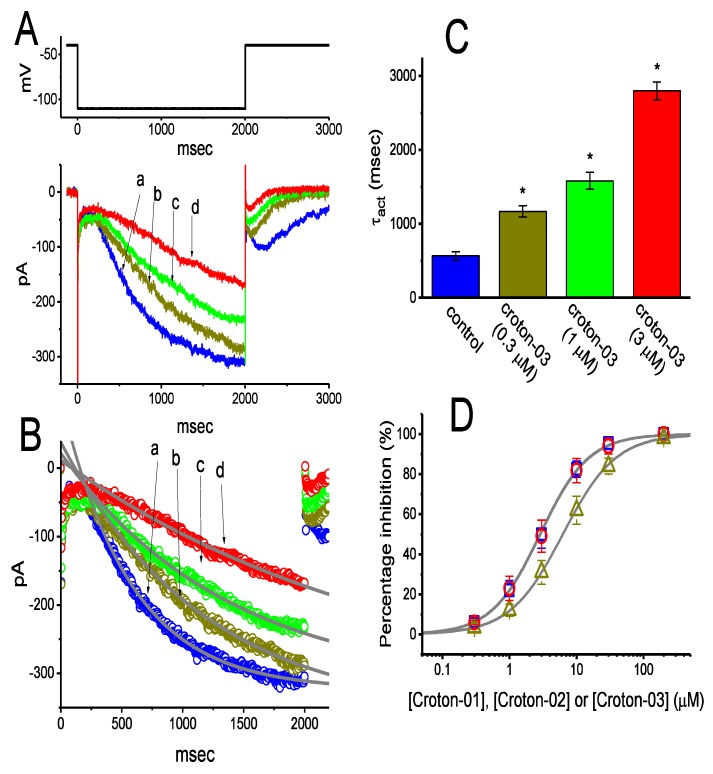
Effects of these *ent*-kaurane-type diterpenoids (i.e., croton-01, croton-02 and croton-03) on hyperpolarization-activated cation current (*I*_h_) recorded from pituitary GH_3_ cells. Cells were bathed in Ca^2+^-free Tyrode’s solution containing 1 μM tetrodotoxin, the recording pipette was filled with K^+^-containing solution, and the examined cell was hyperpolarized from −40 to −110 mV (as indicated in the uppermost part of (A)). (**A**) Representative *I*_h_ traces obtained in the control (a), and during the exposure to 0.3 μM croton-03 (b), 1 μM croton-03 (c), or 3 μM croton-03 (d). (**B**) Current trajectories fitted by single exponential (indicated in the gray line). a: control; b: 0.3 μM croton-03; c: 1 μM croton-03; d: 3 μM croton-03. For better illustrations, data points in each trace were reduced by a factor of 10. (**C**) Summary bar graph showing the effect of croton-03 on the activation time constant (τ_act_) of *I*_h_. The *I*_h_ was evoked by step hyperpolarization from −40 to −100 mV. (**D**) Concentration-dependent inhibition of *I*_h_ by croton-01, croton-02, and croton-03 in GH_3_ cells. The relations between the percentage inhibition of *I*_h_ and the concentrations of croton-01 (O), croton-02 (Δ) or croton-03 (□) are illustrated. Current amplitudes measured at the end of hyperpolarizing step from −40 to −110 mV in the absence or presence of different concentrations of croton-01, croton-02, or croton-03 were compared with the control values. The continuous lines overlaid were fitted by a modified Hill function (see text for details). The IC_50_ values obtained in the presence of croton-01, croton-02, or croton-03 were estimated to be 2.89, 6.25 or 2.84 μM, respectively.

**Figure 2 ijms-21-01268-f002:**
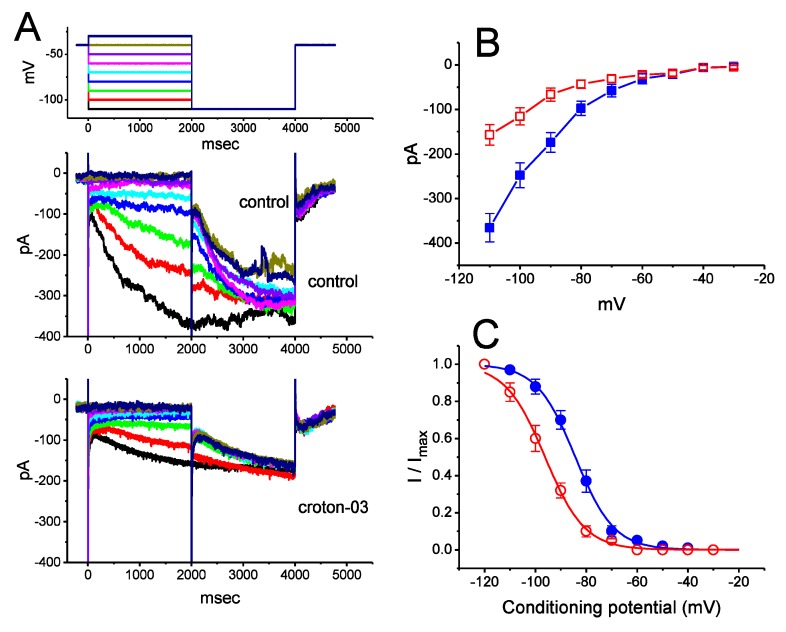
Effect of croton-03 on averaged *I-V* relationship and steady-state activation curve of *I*_h_ in GH_3_ cells. In this set of voltage-clamp current recordings, the conditioning voltage pulses with a duration of 2 sec to potential ranging from −120 and −30 mV. After each conditioning pulse, a test pulse to −110 mV was applied to evoke *I*_h_ for measurement of the steady-state activation curve of the current. The upper part of (A) indicates the voltage protocol applied. An example of current traces obtained by this two-pulse protocol is illustrated in (**A**). (**B**) Averaged *I-V* relationship of *I*_h_ in the control (■) and during the exposure to 3 μM croton-03 (□). Current amplitude was measured at the end of each membrane (i.e., conditioning) potential applied. Each point represents the mean ± SEM (*n* = 7 for each point). (**C**) Effect of croton-03 on the steady-state activation curve of *I*_h_ in the absence (●) and presence (O) of 3 μM croton-03 in GH_3_ cells. The *I*_h_ amplitudes were normalized with those measured at the end of each test pulse and then constructed against the conditioning potentials. The continuous line was least-squares fitted by the Boltzmann equation (see text for details). Please note that the steady-state activation curve of *I*_h_ in GH_3_ cells was significantly shifted toward hyperpolarizing voltage during the exposure to croton-03.

**Figure 3 ijms-21-01268-f003:**
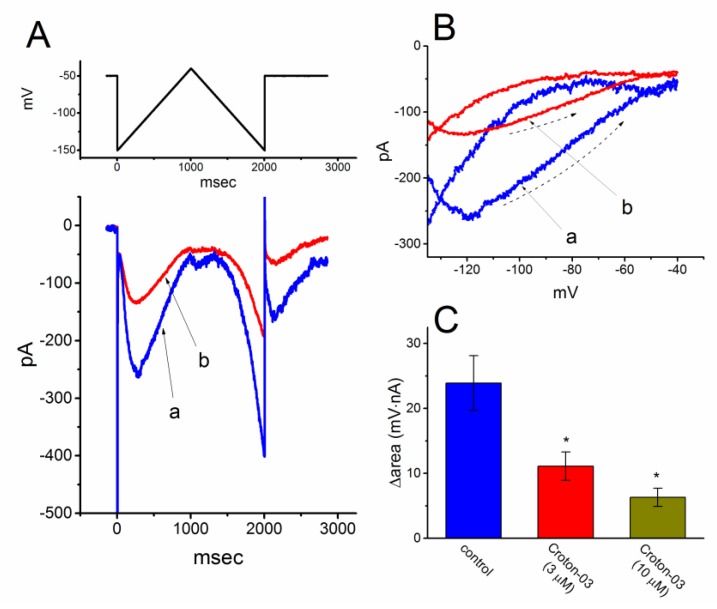
Effect of croton-03 on the voltage hysteresis of *I*_h_ recorded from GH_3_ cells. (**A**) Representative current trace elicited by long-lasting triangular (i.e., upsloping and downsloping) ramp pulse between −150 and −40 mV with a duration of 2 sec. The upper part in (A) is the voltage protocol applied to the examined cell. Current trace labeled a is control and that labeled b was taken in the presence of 3 μM croton-03. (**B**) Voltage hysteresis (i.e., the relationship of forward and reverse current versus membrane voltage) of *I*_h_ measured in the control (a) and during the exposure to 3 μM croton-03 (b). Dashed arrows show the direction of *I*_h_ in which time passes during the elicitation of triangular ramp pulse. (**C**) Summary bar graph showing the effect of croton-03 (3 or 10 μM) on the Δarea of voltage hysteresis of *I*_h_ (mean ± SEM; *n* = 7 for each bar). ^*^ Significantly different from control (*p* < 0.05).

**Figure 4 ijms-21-01268-f004:**
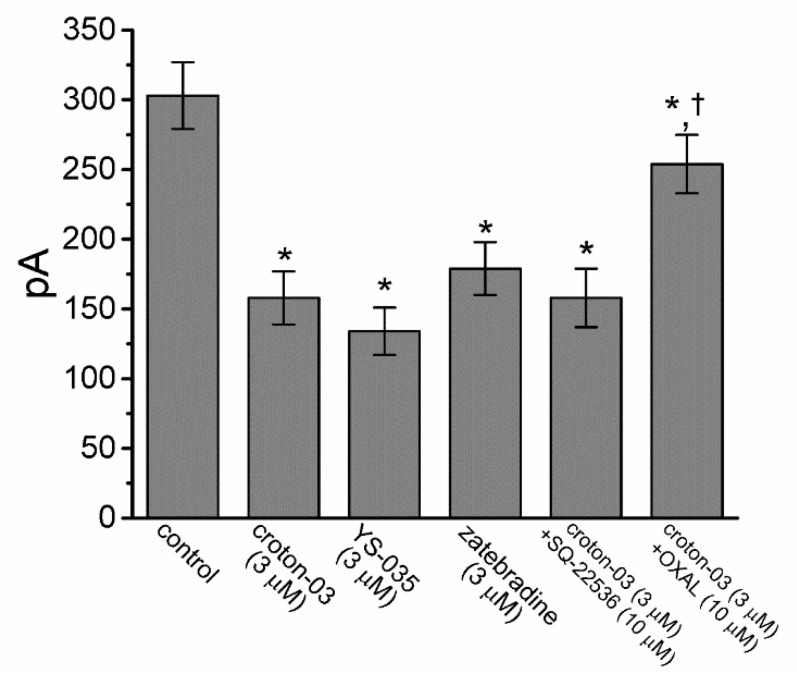
Comparisons among effects of croton-03, YS-035, zatebradine, croton-03 plus SQ-22536 and croton-03 plus oxaliplatin (OXAL) on *I*_h_ amplitude recorded from GH_3_ cells. Cells were bathed in Ca^2+^-free Tyrode’s solution and the hyperpolarizing potential from −40 to −110 mV with a duration of 2 sec was applied to the cell. *I*_h_ amplitude was measured at the end of hyperpolarizing pulse in the presence of different tested compounds. Each bar indicates the mean±SEM (*n* = 7–9). * Significantly different from control (*p* < 0.05) and ^†^ significantly different from croton-03 (3 μM) along group (*p* < 0.05).

**Figure 5 ijms-21-01268-f005:**
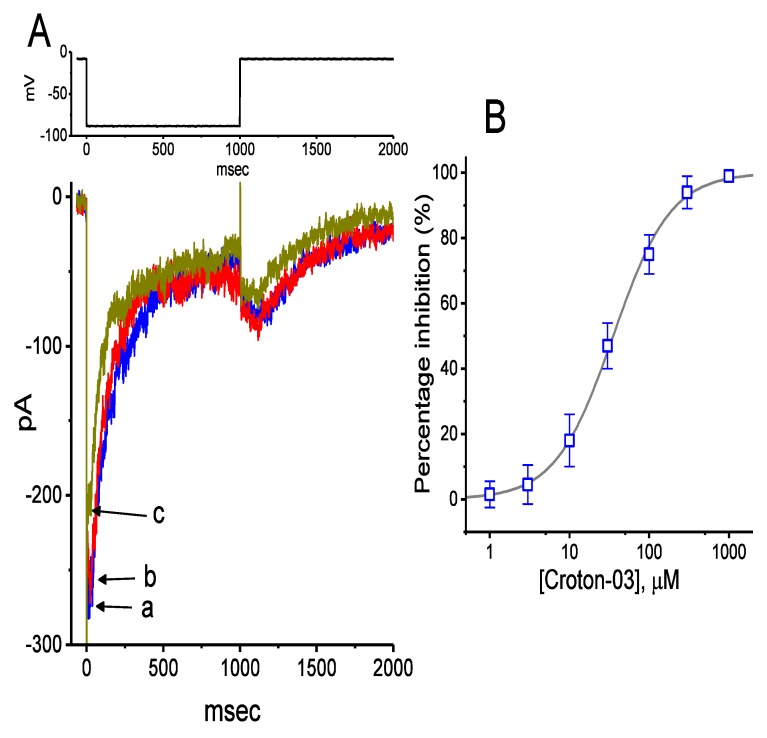
Effect of croton-03 on *erg*-mediated K^+^ current (*I*_K(erg)_) in GH_3_ cells. In these experiments, cells were bathed in Ca^2+^-free, high-K^+^ solution, and the electrode was filled with K^+^-containing solution. (**A**) Representative *I*_K(erg)_ traces obtained in the control (a) and during the exposure to 3 μM croton-03 (b) or 10 μM croton-03 (c). The upper part indicates the voltage protocol applied. (**B**) Concentration-dependent inhibition by croton-03 of *I*_K(erg)_ amplitude (mean ± SEM; *n* = 7–8 for each point). Each cell was hyperpolarized from −10 to −90 mV and current amplitudes obtained during the exposure to different croton-03 concentrations were measured at the beginning of each hyperpolarizing step. The smooth curve was least-squares fitted by the Hill equation (see text for details), and the IC_50_ value was constructed and then estimated to be 35.3 μM.

**Figure 6 ijms-21-01268-f006:**
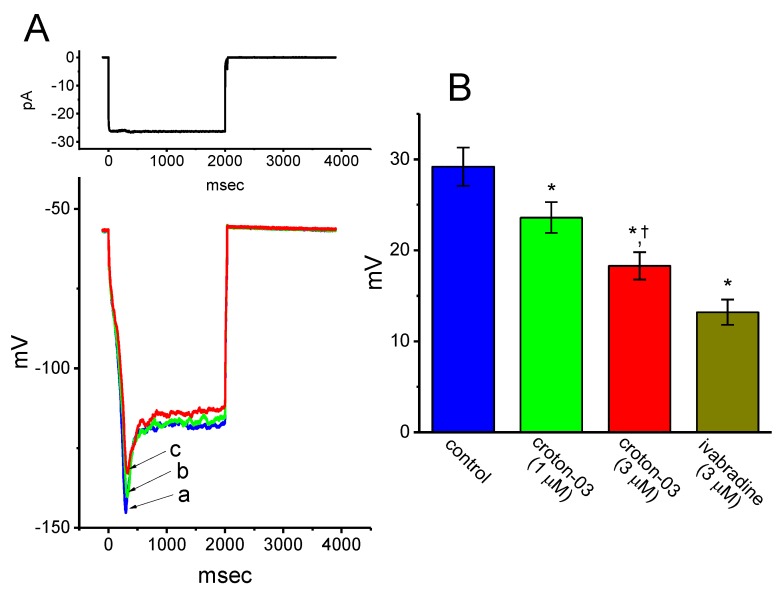
Effect of croton-03 on sag potential identified in GH_3_ cells. Cells were bathed normal Tyrode’ solution containing 1.8 mM CaCl_2_ and 1 μM tetrodotoxin. Current-clamp voltage recordings were made in these experiments and a long-step hyperpolarizing (i.e., inward) current stimulus was applied to the examined cell. (**A**) Superimposed potential traces obtained in the control (a) and during cell exposure to 1 μM croton-03 (b) or 3 μM croton-03 (c). Upper part indicates the long-step current injection applied. (**B**) Summary bar graph showing the effect of croton-03 or ivabradine on the amplitude of sag potential (mean ± SEM; *n* = 6–8 for each bar). The amplitude of sag potential (i.e., difference between the beginning and end of hyperpolarizing stimulus) was taken in the absence and presence of croton-03 or ivabradine. * Significantly different from control (*p* < 0.05) and ^†^ significantly different from croton-03 (1 μM) along group (*p* < 0.05).

**Figure 7 ijms-21-01268-f007:**
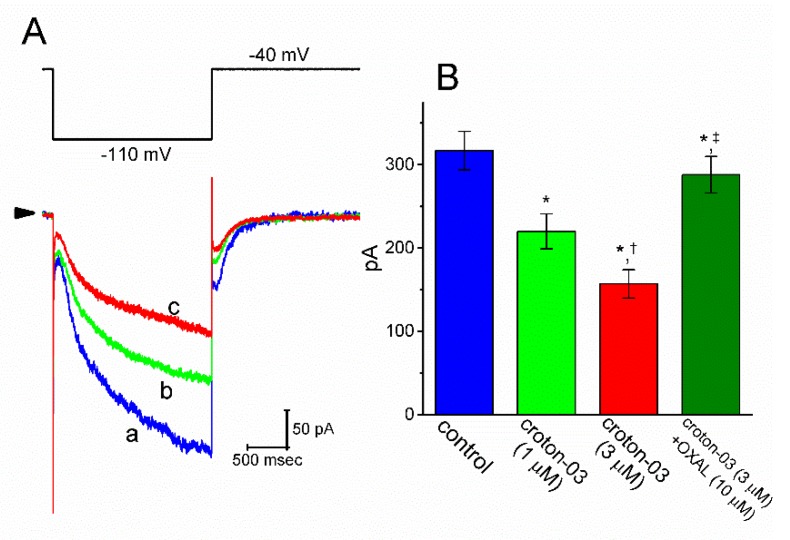
Inhibitory action of croton-03 on *I*_h_ identified in INS-1 cells. This set of experiments was conducted in cells bathed in Ca^2+^-free, Tyrode’s solution and the recording pipette was filled with K^+^-containing solution. (**A**) Representative *I*_h_ traces obtained in the absence (a) and presence of 1 μM croton-03 (b) or 3 μM croton-03 (c). The upper part indicates the voltage protocol applied. (**B**) Summary bar graph showing the effect of croton-03 and croton-03 plus oxaliplatin on *I*_h_ amplitude (mean ± SEM; *n* = 7–8 for each bar). Current amplitude was obtained at the end of 2-sec hyperpolarizing pulse from −10 to −110 mV. OXAL: 10 μM oxaliplatin. * Significantly different from control (*p* < 0.05), ^†^ significantly different from croton-03 (1 μM) along group (*p* < 0.05), and ^‡^ significantly different from croton-03 (3 μM) along group (*p* < 0.05).

**Figure 8 ijms-21-01268-f008:**
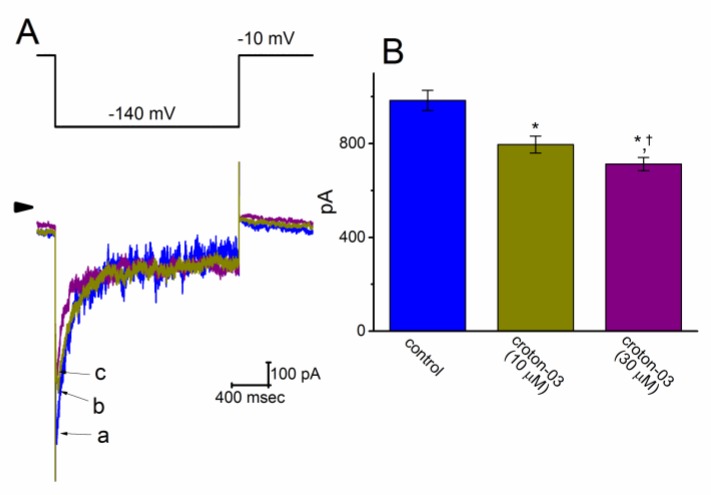
Effect of croton-03 on *I*_K(erg)_ amplitude in INS-1 cells. In these experiments, we bathed in Ca^2+^-free, high-K^+^ solution and, during the measurements, we filled the pipette by using K^+^-containing solution. (**A**) Representative *I*_K(erg)_ traces elicited in response to 1-sec step hyperpolarization from −10 to −140 mV (as indicated in the upper part). The current trace labeled a is control, and those labeled b or c were obtained in the presence of 10 or 30 μM croton-03, respectively. Arrowhead indicates the zero current level and calibration mark shown in the right lower side applies all current traces. (**B**) Summary bar graph of croton-03 effects on the amplitude of deactivating *I*_K(erg)_ elicited by membrane hyperpolarization (mean ± SEM; *n* = 8 for each bar). Current amplitude was taken at the beginning of each hyperpolarizing pulse. * Significantly different from control (*p* < 0.05) and ^†^ significantly different from croton-03 (10 μM) alone group (*p* < 0.05).

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
