# Peer review of "Characterization of Inhibitory Effectiveness in Hyperpolarization-Activated Cation Currents by a Group of ent-Kaurane-Type Diterpenoids from Croton tonkinensis"

_ijms, 2020, doi:10.3390/ijms21041268_

Round 1

Reviewer 1 Report

This manuscript deals with an extensive investigation on the depressive effect of croton-03 upon hyperpolarization–activated cation currents (Ih) stimulated in different cells upon removing calcium ions and injecting potassium ions. Measurements of potential stimulated by a negative current pulse were also carried out, to determine the effect of croton-03 on sag potential at GH3 cells.

1) The terms “suppression” and “suppressive effect” should be removed from the manuscript and replaced by “depression” and “depressive effect”, because in no case is Ih suppression reported. In this respect, the expression “mildly suppressed” on p. 11, line 307, appears as an oxymoron.

2) A comparison between the depressive effects of the three different crotons is only described in Fig. 1D, in connection with Ih in pituitary GH3 cells. This result is too meager to be generalized with detailed quantitative IC50 values on p. 1, line 28; hence, these values should be removed from this line, and only a qualitative statement on the different effect of the three crotons should be made.

3) As a rule, a steady-state inactivation curve of Ih is measured by stepping from evenly spaced holding potentials to a constant more negative potential. The protocol in Fig. 2A refers indeed to this procedure. However, the authors state that “Ih was elicited when the cell was hyperpolarized from –40 mV to a series of voltage steps ranging between –110 add –30 mV”. The authors  should clarify this point by explaining this discrepancy.

4) In Fig. 4, a direct comparison between the depressing effects of croton-03, YS-035 and zatebradine would require a 3 mM concentration of zatebradine.

Minor points:

On p. 3, line 109, letters (c) and (d) are lacking from the legend of Fig. 1A. On p. 4, line 131, and p.11, line 306, the expression “less depolarized potential” is used to denote a “less negative potential”. As a matter of fact, the word “depolarization” refers to an increase in potential; hence, a “less depolarized potential” suggests a more negative potential, rather than a less negative one. Hence, the expression “less depolarized “ potential should be replaced by the much clearer “less negative potential”. In Fig. 3B, the letters ‘a’ and ‘b’ should be shifted to their proper place. On p. 6, line 195, “SQ-22536 is an inhibitor of adenylate cyclase”: Please insert “of”. On p. 6, line 202, it seems to me that “croton-03-mediated DEPRESSION of Ih appears to be independent of AN increase (rather than of ITS increase) in intracellular cyclic AMP”. In Fig. 7, the legend disagrees with the corresponding protocol (–10 mV or –40 mV?).

In my opinion, this manuscript can be accepted for publication in the International Journal of Molecular Sciences after minor revision accounting for this above points.

Author Response

Comments and Suggestions for Authors

This manuscript deals with an extensive investigation on the depressive effect of croton-03 upon hyperpolarization–activated cation currents (Ih) stimulated in different cells upon removing calcium ions and injecting potassium ions. Measurements of potential stimulated by a negative current pulse were also carried out, to determine the effect of croton-03 on sag potential at GH3 cells.

Ans: Thanks for the reviewer’s comments.

The terms “suppression” and “suppressive effect” should be removed from the manuscript and replaced by “depression” and “depressive effect”, because in no case is Ih suppression reported. In this respect, the expression “mildly suppressed” on p. 11, line 307, appears as an oxymoron.

Ans: As per the suggestion by the reviewer, “suppression” or “suppressive effect” appearing throughout the text in the original manuscript was replaced with “depression” or “depressive effect”.  In line 316 of the revised manuscript, “mildly suppressed” was replaced with “mildly depressed” as well.

A comparison between the depressive effects of the three different crotons is only described in Fig. 1D, in connection with Ih in pituitary GH3 This result is too meager to be generalized with detailed quantitative IC50 values on p. 1, line 28; hence, these values should be removed from this line, and only a qualitative statement on the different effect of the three crotons should be made.

Ans: Thanks for the comment raised by the reviewer, the sentence was accordingly changed (lines 27-28 in the revised manuscript).

As a rule, a steady-state inactivation curve of Ih is measured by stepping from evenly spaced holding potentials to a constant more negative potential. The protocol in Fig. 2A refers indeed to this procedure. However, the authors state that “Ih was elicited when the cell was hyperpolarized from –40 mV to a series of voltage steps ranging between –110 add –30 mV”. The authors  should clarify this point by explaining this discrepancy.

Ans: Thanks for bringing our attention. Hence, the legend in the Figure 2 was appropriately revised (lines 135-148 in the revised manuscript). Moreover, the inactivation curve of Ih in the original manuscript was appropriately replaced with “activation” curve of the current appearing throughout the text in the revised manuscript. Figure 2C was hence redone.

4) In Fig. 4, a direct comparison between the depressing effects of croton-03, YS-035 and zatebradine would require a 3 mM concentration of zatebradine.

Ans: As advised by the reviewer, the zatebradine concentration used at 3 microM was replaced in the bar graph of Figure 4 appearing in the revised manuscript. Figure 4 was redone.

Minor points:

On p. 3, line 109, letters (c) and (d) are lacking from the legend of Fig. 1A. On p. 4, line 131, and p.11, line 306, the expression “less depolarized potential” is used to denote a “less negative potential”. As a matter of fact, the word “depolarization” refers to an increase in potential; hence, a “less depolarized potential” suggests a more negative potential, rather than a less negative one. Hence, the expression “less depolarized “ potential should be replaced by the much clearer “less negative potential”. In Fig. 3B, the letters ‘a’ and ‘b’ should be shifted to their proper place. On p. 6, line 195, “SQ-22536 is an inhibitor of adenylate cyclase”: Please insert “of”. On p. 6, line 202, it seems to me that “croton-03-mediated DEPRESSION of Ih appears to be independent of AN increase (rather than of ITS increase) in intracellular cyclic AMP”. In Fig. 7, the legend disagrees with the corresponding protocol (–10 mV or –40 mV?).

Ans:

Goof! We made a mistake. Letters (c) and (d) were included in the revised manuscript (line 110 in the revised manuscript). Thanks for the reviewer’s comments, in lines 132 and 314 of the revised manuscript, “a more negative potential” was hence corrected.

In my opinion, this manuscript can be accepted for publication in the International Journal of Molecular Sciences after minor revision accounting for this above points.

Ans: Thanks!

Reviewer 2 Report

The paper is essentially an electrophysiological characterization of the effects of three similar diterpenoids on HCN and ERG currents in two related cell models (pituitary tumor and insulin-secreting cells). The study seems to have been conducted carefully, and I have only minor comments. However, I object to the notion that the results may have a real physiological, nutrition-related relevance. This is because the IC50 the authors determine is 2.84 micromolar in the best of cases, and I sincerely doubt that levels approaching this value may be reached anywhere in the organism upon consumption of even large amounts of Croton tonkinensis. At least I would need some evidence on this point. One can in principle isolate the compounds or make them chemically, and ingest plenty of the stuff, but the molecules (their structure is not shown, it should be even if they have been already published) are complicated.

Aside from this, the main point I have is that I do not really understand the finding that hysteresis increased as the ramping rate decreased (p. 5, lines 159-160). At first sight this is surprising – despite previous reports that HCN channels display two modes (e.g. Männikkö R, Pandey S et al., J Gen Physiol, 2005) – and needs to be discussed. Furthermore, I do not see evidence of an effect of croton-03 on this behaviour. In the exemplary data provided in p. 5 lines 162-167 the reduction of current due to hysteresis upon return to -100 mV is about 45% without croton-03 and about 43% with it. It is also unsurprising, if one looks at the curves in Fig. 3B, that the area between them decreases in the presence of the inhibitor. This is just a consequence of the dampening of current conduction throughout, and hardly worth mentioning. So, the whole part concerning hysteresis needs clarification/revision.

Section 4.1, lines 377-382: not clear how many fractions were taken; there seem to have been sub-fractions

Language is more or less OK, but there are some points that need adjusting. Examples:

line 66: encoded by, not from

line 81: “the possible modifications of croton-03 on Ih” ? Presumably: the modifications of Ih induced/caused by croton-03

line 269: “.. could significantly attenuate the Ih amplitude suppressed by croton-03”.   ? Perhaps suppression instead of suppressed?

The 2nd paragraph in Discussion is repetitive

line 324: became slower (?)

line 345: could conceivably be an important target

lines 355-358: move to introduction

Author Response

Comments and Suggestions for Authors

The paper is essentially an electrophysiological characterization of the effects of three similar diterpenoids on HCN and ERG currents in two related cell models (pituitary tumor and insulin-secreting cells). The study seems to have been conducted carefully, and I have only minor comments. However, I object to the notion that the results may have a real physiological, nutrition-related relevance. This is because the IC50 the authors determine is 2.84 micromolar in the best of cases, and I sincerely doubt that levels approaching this value may be reached anywhere in the organism upon consumption of even large amounts of Croton tonkinensis. At least I would need some evidence on this point. One can in principle isolate the compounds or make them chemically, and ingest plenty of the stuff, but the molecules (their structure is not shown, it should be even if they have been already published) are complicated.

Ans: Thank for the reviewer’s comment. An additional paragraph regarding this issue was hence included in the Discussion section of the revised manuscript (lines 354-358).

Aside from this, the main point I have is that I do not really understand the finding that hysteresis increased as the ramping rate decreased (p. 5, lines 159-160). At first sight this is surprising – despite previous reports that HCN channels display two modes (e.g. Männikkö R, Pandey S et al., J Gen Physiol, 2005) – and needs to be discussed. Furthermore, I do not see evidence of an effect of croton-03 on this behaviour. In the exemplary data provided in p. 5 lines 162-167 the reduction of current due to hysteresis upon return to -100 mV is about 45% without croton-03 and about 43% with it. It is also unsurprising, if one looks at the curves in Fig. 3B, that the area between them decreases in the presence of the inhibitor. This is just a consequence of the dampening of current conduction throughout, and hardly worth mentioning. So, the whole part concerning hysteresis needs clarification/revision.

Ans: Thanks for the reviewer’s comments. Indeed, the present observations were consistent with previous studies (Manniko et al., 2005; Furst and D’Avanzo, 2015). In order to reduce the concern of the reviewer, additional statements were included in the revised manuscript (lines 169-171). That is, “However, the magnitude of croton-03-induced current inhibition at the upsloping and downsloping limbs of triangular ramp did not differ significantly.  That is, the presence of 3 mM croton decreased Ih amplitude (at -100 mV) in the upsloping and downsloping limb by about 40%.”. Moreover, for clarity, additional results were also included in the revised manuscript (lines 174-176). That is, “For example, as the duration of triangular ramp was prolonged from 2 to 3 sec, the area under the curve between forward and backward current traces (i.e., Darea) increased from 23.9±.4.2 to 34.1±4.9 mV·pA, (n=7, P<0.05).”. Moreover, another sentence was changed to “For example, apart from its depression of Ih magnitude, addition of croton-03 (3 mM) significantly decreased the area by about 50% elicited in response to such long-lasting triangular voltage ramp, as evidenced by a significant decrease of Darea from 23.9±.4.2 to 11.1±2.2 mV·nA (n=7, P<0.05)” (lines 178-181).

Section 4.1, lines 377-382: not clear how many fractions were taken; there seem to have been sub-fractions

Ans: Extraction and fractionation have been described previously (Kuo et al., J Nat Prod 2007;70(12):190609). The n-hexane soluble fraction was subjected to silica gel column chromatograph eluted with a step gradient to afford nineteen fractions (H1-H19).

Language is more or less OK, but there are some points that need adjusting.

Ans: Thanks for the comments.

Examples:

line 66: encoded by, not from

Ans: “encoded by” was changed (line 65 in the revised manuscript), as the reviewer pointed out.

line 81: “the possible modifications of croton-03 on Ih” ? Presumably: the modifications of Ih induced/caused by croton-03

Ans: The sentence was changed to “…. the modifications of Ih caused by croton-03…” (line 82 in the revised manuscript).

line 269: “.. could significantly attenuate the Ih amplitude suppressed by croton-03”.   ? Perhaps suppression instead of suppressed?

Ans: The sentence was changed to “… attenuate croton-03-mediated inhibition of the Ih amplitude.” In lines 276-277 of the revised manuscript.

The 2nd paragraph in Discussion is repetitive

Ans: Thanks for the reviewer’s comments, the 2nd paragraph in the Discussion section was accordingly removed in the Discussion section of the revised manuscript.

line 324: became slower (?)

Ans: “became slowed” was corrected (line 327 of the revised manuscript).

line 345: could conceivably be an important target

Ans: Thanks for bringing our attention. “…could be conceivably be an important target…” was corrected (line 347 of the revised manuscript).

lines 355-358: move to introduction

Ans: As advised by the reviewer, the sentence was moved to the Introduction section of the revised manuscript (lines 76-78). The reference number was accordingly corrected.